# Interplay of protein corona and immune cells controls blood residency of liposomes

Francesca Giulimondi[1,6], Luca Digiacomo[1,6], Daniela Pozzi[1], Sara Palchetti[1], Elisabetta Vulpis[1], Anna Laura Capriotti[2], Riccardo Zenezini Chiozzi [2], Aldo Laganà[2], Heinz Amenitsch [3], Laura Masuelli[4], Morteza Mahmoudi [5], Isabella Screpanti[1], Alessandra Zingoni[1] & Giulio Caracciolo[1]

In vivo liposomes, like other types of nanoparticles, acquire a totally new 'biological identity' due to the formation of a biomolecular coating known as the protein corona that depends on and modifies the liposomes' synthetic identity. The liposome–protein corona is a dynamic interface that regulates the interaction of liposomes with the physiological environment. Here we show that the biological identity of liposomes is clearly linked to their sequestration from peripheral blood mononuclear cells (PBMCs) of healthy donors that ultimately leads to removal from the bloodstream. Pre-coating liposomes with an artificial corona made of human plasma proteins drastically reduces capture by circulating leukocytes in whole blood and may be an effective strategy to enable prolonged circulation in vivo. We conclude with a critical assessment of the key concepts of liposome technology that need to be reviewed for its definitive clinical translation.

[1] Department of Molecular Medicine, Sapienza University of Rome, Viale Regina Elena 291, 00161 Rome, Italy. [2] Department of Chemistry, Sapienza University of Rome, P.le Aldo Moro 5, 00185 Rome, Italy. [3] Institute of inorganic Chemistry, Graz University of Technology, Stremayerg 6/IV, 8010 Graz, Austria. [4] Department of Experimental Medicine, Sapienza University of Rome, Viale Regina Elena 324, 00161 Rome, Italy. [5] Precision Health Program, Michigan State University, East Lansing, MI 48823, USA. [6] These authors contributed equally: Francesca Giulimondi, Luca Digiacomo. Correspondence and requests for materials should be addressed to M.M. (email: mahmou22@msu.edu) or to G.C. (email: giulio.caracciolo@uniroma1.it)

Over the last decades, liposomes have been widely characterized and their application is now well established in various areas, such as drug and gene delivery[1]. In these fields, liposomes are one of the most versatile colloidal carrier systems, since they have structural features that allow the encapsulation, protection, and transportation of molecules with different physical–chemical properties[2–4].

The molecular composition of these formulations of liposomes can be finely tuned according to the properties of the active compound and the intended application. Important factor in this context is the surface charge type (e.g., cationic charge). Due to favorable electrostatic interactions, cationic liposomes (CLs) form stable complexes with negatively charged nucleic acids (NAs)[5]. CL-DNA complexes (lipoplexes) are the most promising nonviral nanocarriers in several gene-delivery applications, such as gene therapy and gene vaccination[6,7]. Neutrally charged liposomes (NLs) are made of zwitterionic lipids (e.g., phosphatidylcholine, cholesterol etc.), and are typically used to deliver anticancer drugs (e.g., doxorubicin, paclitaxel etc.)[8]. Anionic liposomes (ALs) are preferentially used for drug delivery[8], but in combination with cations (e.g., $Ca^{2+}$, $Mg^{2+}$, $Fe^{3+}$ etc.) can also form lipoplexes working therefore as gene-delivery systems[9].

Despite the ease of preparation and functionalization of liposomes[10], their clinical application is associated with numerous drawbacks[11]. More than three decades of drug-delivery research have correlated opsonization by human plasma (HP) proteins with uptake of liposomes by resident macrophages and fast removal from bloodstream[12,13]. It was demonstrated that particle sequestration by circulating leukocytes is a major barrier to drug delivering, especially with non-PEGylated nanoparticle formulations[14]. Grafting PEG to the liposome surface has long been considered an efficient strategy to limit opsonisation[15], but recent findings clarified that PEGylation cannot fully prevent the absorption of plasma proteins at the liposome surface[16]. A paradigm shift in drug delivery has been proposed, i.e., that exposure of liposomes to biological fluids leads to formation of a protein interface termed protein corona (PC)[17–19] that controls circulation lifetime, drug release profile, biodistribution, nanotoxicity, and cellular interactions of liposomes in vivo[20,21]. The structure and composition of PC depends on the liposomes' synthetic identity (size, surface charge, lipid composition)[22], the protein source (e.g., human plasma vs. mouse plasma[23]), and environmental factors (e.g., temperature[24] and duration of exposure[19,25]). Previous studies demonstrated that PC can recruit specific proteins from plasma[26], which eventually prevent particle recognition by macrophages[27,28].

Thus, the major aspect of our work focuses on the question, whether the pre-adsorbed corona proteins may allow liposomes to avoid capture by circulating leukocytes, when they are exposed to patient's whole blood. This question needs to be addressed in order to reveal if liposomes pre-coating may be an effective strategy to enable prolonged circulation in vivo. However, it has been demonstrated that plasma protein concentration is a major factor affecting biological identity of nanoparticles[29]. Therefore, mechanistic understanding of liposomes' biological identity as a function of protein concentration is a necessary step to engineer pre-coated liposomes with proper physico-chemical properties (e.g., size and surface charge) and PC composition. As model systems of CLs, NLs`, and ALs, we employ liposomes made of cationic 1,2-dioleoyl-3-trimethylammonium-propane (DOTAP), neutral dioleoylphosphocholine (DOPC), and anionic 1,2-dioleoyl-sn-glycero-3-phospho-(1′-rac-glycerol) (DOPG). Recent studies demonstrated that PC formed in vivo[30–33] may be different from its counterpart formed under static incubation in vitro. This is in agreement with former findings showing that shear stress produced by circulating fluids affects the PC structure and composition[34,35]. However, in this work, liposomes are exposed to static HP since pre-coating is aimed at creating an artificial PC and not to reproduce a PC mimetic of that which is formed in vivo.

Here, we demonstrate that the protein corona controls the interaction of liposomes with immune cells. We also reveal that the pre-coating liposomes with an artificial corona made of human plasma proteins significantly decreases capture by circulating leukocytes in whole blood, thus, representing an efficient strategy to escape sequestration by immune cells and enable prolonged circulation in vivo.

## Results

**Size and zeta-potential results.** Preliminary experiments were aimed at characterizing zeta potential and hydrodynamic diameter, $D_H$, of pristine DOTAP, DOPC, and DOPG (Supplementary Table 1). Next, we exposed DOTAP, DOPC, and DOPG to increasing protein concentration from HP = 1% to HP = 50%. Figure 1a shows zeta potential of DOTAP, DOPC, and DOPG as a function of HP percentage. Zeta potential of DOTAP becomes neutral at 5% < HP < 10%, and shifts toward negative values as the concentration of plasma increases. DOPC liposomes are neutrally charged, and their zeta potential monotonously decreases with increasing HP percentage. Pristine DOPG vesicles are highly anionic and zeta potential monotonously increases as a function of plasma concentration. It is noteworthy that the zeta-potential profiles of DOTAP, DOPC, and DOPG follow a monotonic trend with increasing HP converging to a common negative plateau value (zeta potential ~ −23 mV) that is independent on the pristine liposome surface charge. Such "normalization" of zeta potential is produced by surface adsorption of plasma proteins and is in good agreement with previous findings[36].

Next, we explored the effect of PC on the liposome size. As shown in Fig. 1b, size of DOTAP increases at low plasma concentration, exhibits a local maximum in correspondence to charge inversion (i.e., 5%<HP<10%), and it decreases at higher plasma content (HP>10%). This behavior resembles the typical "re-entrant condensation" of CLs in the presence of nucleic acids (NAs)[5]. Formation of large-size DOTAP–protein complexes occurs around the "isoelectric point", i.e., when positive charge of DOTAP vesicles and negative charge of plasma proteins balance each other. When this occurs, the van der Waals short-range attraction dominate over Coulomb electrostatic repulsion thus promoting formation of large aggregates[5,37]. As the protein content increases, electrostatic repulsion returns to prevail leading to the formation of decreasing-size complexes until the starting size is reached again. The evolution of zeta potential and size of DOTAP–protein complexes suggests that plasma proteins act as a molecular glue between distinct DOTAP vesicles and that the protein concentration is one of the key factors controlling the equilibrium structure of CL–protein complexes. The implication of these observations is relevant, as clearance cells recognize objects larger than (roughly) 300 nm and remove them from the bloodstream[38]. Evolution trend is much less prominent for DOPC, while it is not detected for DOPG (insets of Fig. 1b). Finally, at high plasma concentration, all the formulations tend to assume similar size. In summary, here, we show that (i) protein concentration is a key factor affecting zeta potential and size of pre-coated liposomes; (ii) at low protein concentration, size and zeta potential of pre-coated liposomes are strongly affected by surface charge type of pristine vesicles; (iii) at high protein concentration, liposome–protein complexes exhibit very similar size and zeta potential. All these results are in good agreement with previous results obtained using other nanoparticles of

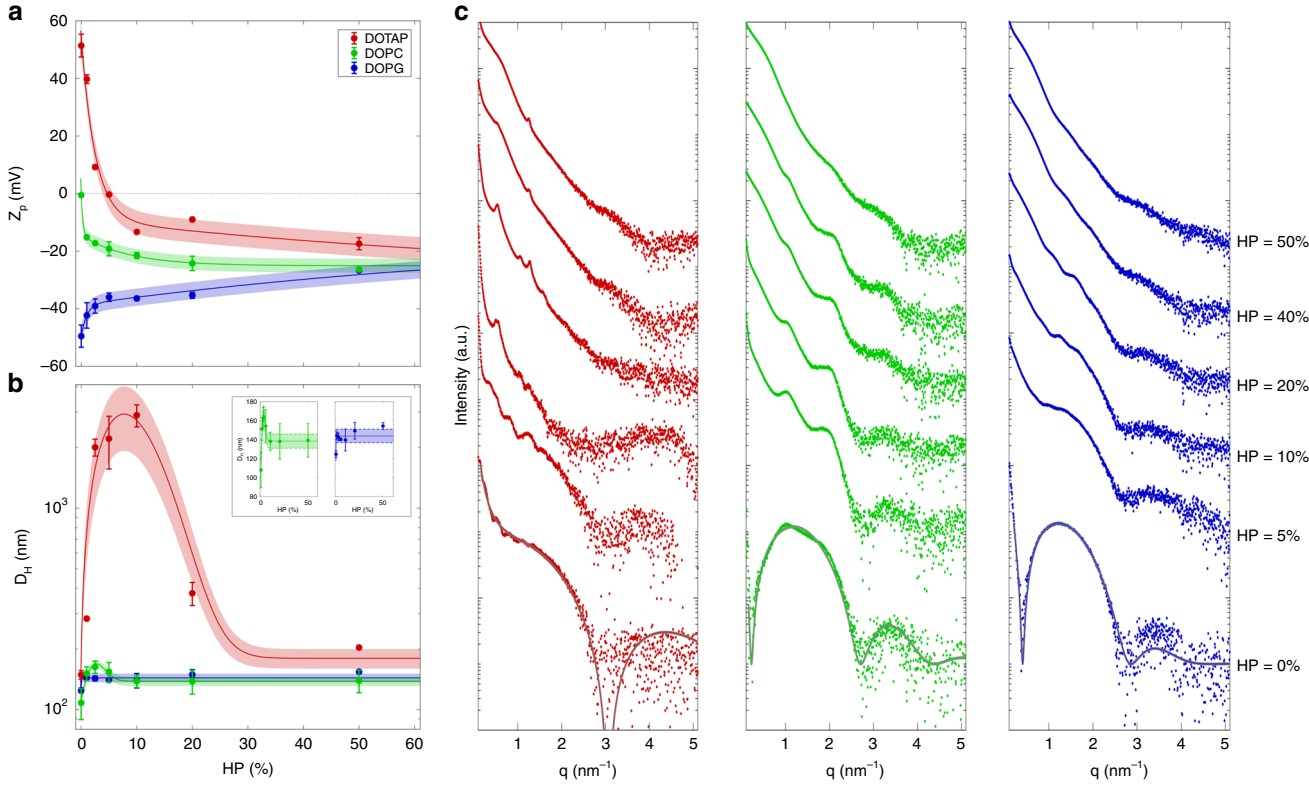

**Fig. 1** Characterization of liposome–protein complexes. Zeta-potential ($Zp$, panel **a**) and hydrodynamic diameter ($D_H$, panel **b**) of cationic DOTAP (red), neutral DOPC (green), and anionic DOPG (blue) liposomes as functions of human plasma (HP) concentration. For clarity, size of DOPC-HP and DOPG-HP complexes are reported also in the inset of panel **b**. **c** Synchrotron small angle X-ray scattering patterns of liposome–protein complexes as a function HP concentration. The results are given as mean ± standard deviation of three replicates

different materials. In particular, Cukalevski et al. showed that low protein concentrations promoted formation of large clusters of polystyrene and polymethyl methacrylate nanoparticles, whereas high protein concentrations induced small-size aggregates[39].

**Nanostructure results.** Next, we addressed the question whether pre-coating changed the nanostructure of liposomes. This is a key issue, since the nanostructure influences the intracellular fate of liposomes[40] and structural changes (e.g., vesicle aggregation and/or rupture) could affect drug/gene bioavailability[41]. To evaluate the impact of protein adsorption on the inner structure of DOTAP, DOPC, and DOPG, we carried out high-resolution synchrotron SAXS experiments. Figure 1c shows the measured SAXS patterns for the three investigated formulations at different HP concentrations. Bottom curves represent the SAXS patterns of bare liposomes (HP = 0%). Exposure of DOTAP to low plasma concentration induces formation of multilamellar structure as demonstrated by the presence of Bragg peaks. According to the literature[42], we propose that plasma proteins coat the surface of liposomes leading to clusters of protein-decorated vesicles. The anisotropic stresses generated by protein-mediated liposome–liposome adhesion may cause local rupture of vesicles leading to the growth of multilamellar particles. As previously demonstrated for liposome–DNA complexes[42], formation of clusters can start only when the external surface of liposomes has been completely coated by plasma proteins. We also observe that the intensity of Bragg peaks decreases with increasing HP percentage. This effect is due to the increasing amount of unbound proteins whose scattering signal masks Bragg peaks. At high plasma concentration, no plain evidence of multi-lamellar structure is found. According to zeta-potential results (Fig. 1a), this may be due to massive protein binding to liposome

surface that creates a repulsive barrier between liposomes. A much less evident process takes place when DOPC is exposed to HP. On the other side, incubation of DOPG in HP does not originate multilamellar complexes (Fig. 1c). SAXS results let us conclude that nanostructure of pre-coated liposomes is affected by surface charge type of bare liposomes at low protein concentration. On the other side, nanostructure of pre-coated liposomes is independent from synthetic identity of pristine vesicles at high plasma concentration.

**Protein-binding results.** Next, we asked whether the physico-chemical properties of liposome–protein complexes correlated with the amount of proteins bound to liposomes. Supplementary Table 2 shows the nanograms of proteins/milligram of lipid absorbed onto the surface of DOTAP, DOPC, and DOPG liposomes as a function of protein concentration. At low protein concentration, DOTAP adsorbs large amount of proteins that increases with HP percentage. This finding is in agreement with DLS, zeta potential and SAXS results showing formation of large-size aggregates. Protein bound to DOPC and DOPG is much less than that adsorbed to DOTAP and increases with increasing plasma concentration.

As above specified, recovery of liposomes from the blood[31,43] or separation of liposomes from plasma incubation in vitro has proven that liposomes are coated by plasma proteins. As a consequence of protein coating, liposomes lose synthetic identity and the PC is the new relevant bio-nano interface that interacts with biological systems[44].

**Composition of the liposome–protein corona.** Another key factor affecting biological identity of liposomes is therefore composition of PC. As a next step, we characterized composition

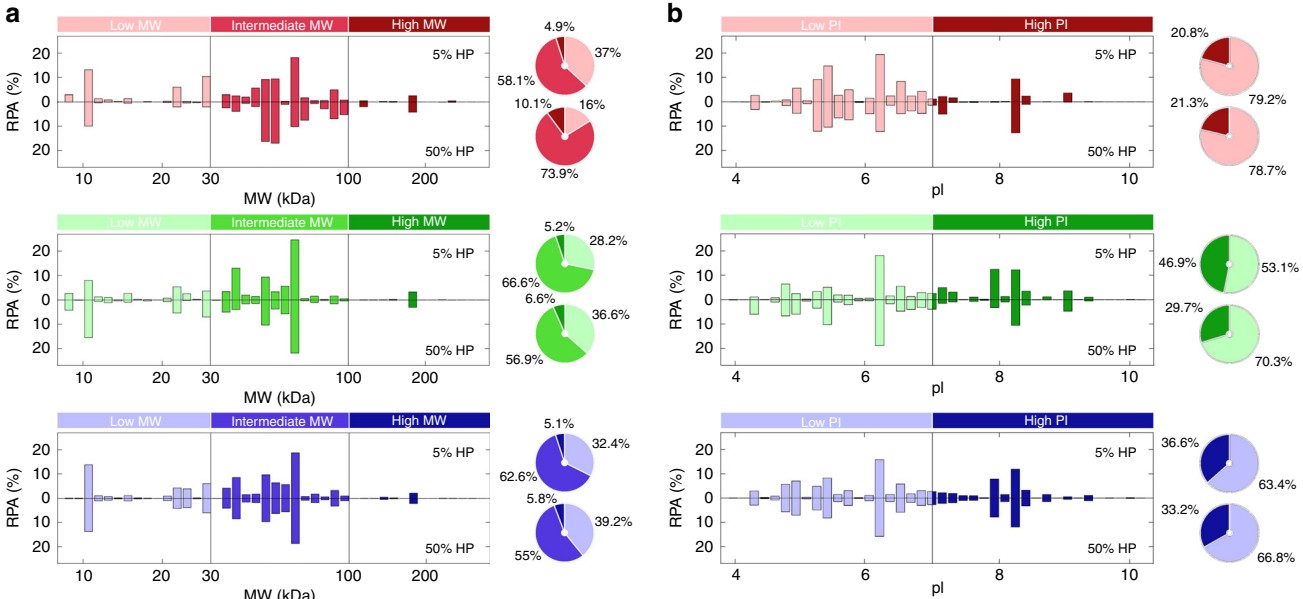

**Fig. 2** Classification of corona by molecular mass and isoelectric point. Relative protein abundance (RPA) of the corona proteins identified on liposomes by nanoLC–MS/MS after interaction with human plasma (HP) according to their molecular weight (**a**) and calculated isoelectric point (**b**) for DOTAP (red), DOPC (green), and DOPG (blue) liposomes. Each liposome formulation was incubated with HP = 5 % (upper panels) and HP = 50% (lower panels)

of PCs at low (HP = 5%) and high (HP = 50%) protein concentration, where the largest difference in physico-chemical properties of liposome–protein complexes were found. Supplementary Table 3 lists all the proteins identified in the respective liposomes' coronas by nano-LC-MS/MS. Categorizing proteins according to molecular weight (MW) allowed us to reveal that plasma proteins with intermediate MW (30–100 kDa) are the largest fraction of the PC (Fig. 2a). We also observe that proteins with isoelectric point (pI) <7 at physiological pH are the main component for all liposomes, irrespective of the liposomes' synthetic identity or plasma concentration (Fig. 2b). These results are in agreement with those reported by Tenzer et al.[19] for inorganic nanoparticles. Supplementary Table 4 lists the relative protein abundance (RPA) of plasma proteins grouped according to biological processes of the blood system. MS/MS results show that the PC of DOPC is poorly affected by protein percentage, while those of DOTAP and DOPG present significant evolution as a function of increasing protein concentration. In particular, exposing DOTAP to increasing protein concentration promotes the diminution in the RPA of acute phase, immunoglobulins, and lipoproteins, as well as the increase in the concentration of coagulation proteins. According to recent findings[45], diminution in immunoglobulins may be relevant as it is related to lower complement activation efficiency. Minor changes are observed for complement and tissue leakage proteins. When DOPG is exposed to increasing plasma concentration, we observe a simultaneous reduction in the RPA of immunoglobulins, coagulation, complement, and tissue leakage proteins that is balanced by massive accumulation of lipoproteins whose RPA boosts from 28.5% at HP = 5% to 45.3% at HP = 50%. Figure 3 shows detailed values for individual proteins within each functional class. We observe that A1AT, HPT, IGG1, IGK, IGL1, and APOA1 account for the largest variation in the PC of DOTAP. Protein concentration significantly affects the enrichment of FIBA, FIBB, FIBG, C1QA, C1QB, C1QC, CO3, APOA1, APOC2, APOC3, APOH, K1C9, K1C10, and C21 in the PC of DOPG. Other relevant individual proteins are reported in Supplementary Table 5. The most significant changes are found for HSA, clusterin, inter-alpha antitrypsin, and vitronectin in DOTAP–HP complexes.

Globally, the results reported in Figs. 1–4 show that size, surface charge, nanostructure, and PC composition of pre-coated liposomes at HP = 50% are optimal for drug-delivery applications, and were therefore used in the following experiments aimed at correlating biological identity of liposomes with their physiological response.

**Cellular uptake by monocytic THP1 cells.** The main concern associated with liposome-mediated drug delivery is no doubt fast removal from bloodstream. Macrophages are the first line of defense against invading foreign agents, including liposomes[46], and how macrophages interact with liposomes determines particle clearance. As a first step, cellular uptake of pre-coated liposomes is evaluated using human monocytic THP1 cells by flow cytometry. In the absence of PC, cellular uptake of DOTAP was much higher than that of DOPC and DOPG. This is likely due to its cationic charge that promotes not specific cellular association by electrostatic attraction with negatively charged plasma membrane. At low plasma concentration (HP = 5%), capture of liposome–protein complexes by THP1 human monocytic cell line was highly dependent on charge type with cellular uptake in the order: DOTAP > DOPG > DOPC. For DOTAP–protein complexes, median fluorescence intensity increases with increasing protein concentration, is maximum at HP = 5%, then it monotonously decreases with increasing protein concentration (Fig. 4a). At low protein concentration, cellular uptake of DOPC–protein and DOPG–protein complexes was much lower than that of DOTAP–protein ones. At high protein concentration (HP = 50%), cellular uptake was low, irrespective of the synthetic identity of pristine liposomes. This would suggest that in vitro results at high plasma concentration do not match most clearance results reported in vivo[12,43,47]. This disputes an accepted paradigm in drug-delivery research, i.e., that in vitro studies have to be performed at high plasma concentration to be mimetic of in vivo conditions. We therefore suggest the need for a re-evaluation of how in vitro studies should be planned in the future and how in vitro–in vivo extrapolations can be made.

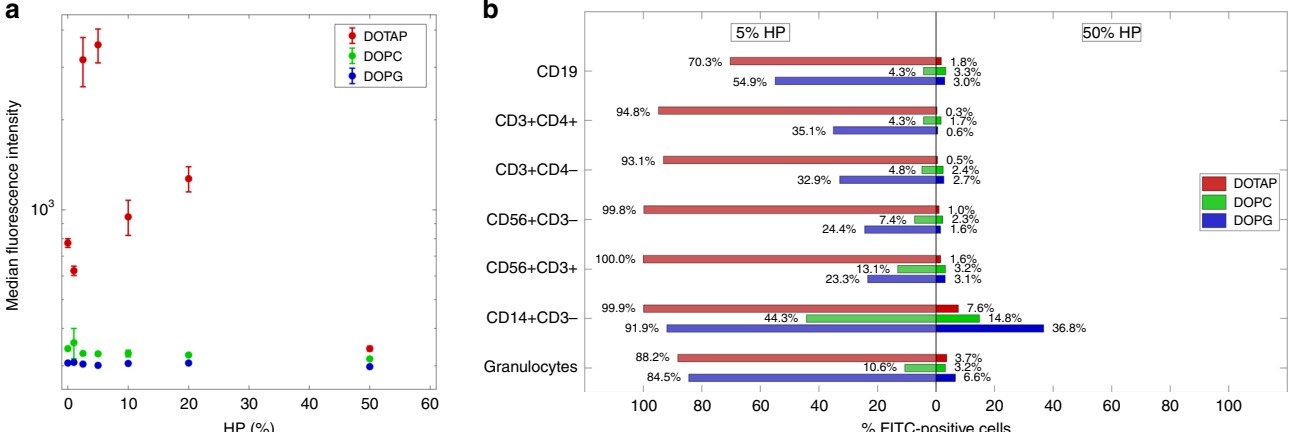

**Fig. 3** Bioinformatic classification of corona proteins. Relative protein abundance (RPA) of plasma proteins identified in the respective liposomes' coronas by quantitative nanoLC-MS/MS at the indicated human plasma concentrations. Plasma proteins were grouped according to functional processes: acute phase (**a**), coagulation (**b**), complement (**c**), immunoglobulins (**d**), lipoproteins (**e**), tissue leakage (**f**). Each value is the average of triplicates ± standard deviation within a single experiment

**Fig. 4** Sequestration of liposome–protein complexes by immune cells in vitro. **a** Cellular uptake of fluorescently labeled liposome–protein complexes by human monocyte THP1 cells via flow cytometry: DOTAP (red), DOPC (green), and DOPG (blue). Complexes were prepared at several human plasma (HP) concentrations: 0 (pristine liposomes), 1%, 2.5%, 5%, 10%, 20%, and 50%. Each value is the average of triplicate samples ± standard deviation within a single experiment. **b** Leukocyte uptake of liposome–protein complexes via flow cytometry: DOTAP (red), DOPC (green), and DOPG (blue). Complexes were prepared at low (5%) and high (50%) human plasma (HP) concentrations. Each value is the average of duplicate samples ± standard deviation within a single experiment. The fluorescence of internalized liposomes was measured as the percentage of FITC-positive cells by gating on distinct leukocyte subpopulations as indicated. The gating strategy was obtained as shown in Supplementary Fig. 3

**Particle sequestration by leukocytes in patients' whole blood.** Following step is evaluating sequestration by distinct leukocyte subpopulations derived from peripheral blood mononuclear cells (PBMCs) of healthy donors. Pre-coated DOTAP prepared at low protein concentration (HP = 5%) are avidly internalized by almost all the populations analyzed (Fig. 4b). On the other side, sequestration of pre-coated DOPC and DOPG by PBMCs was lower that than of DOTAP and, for each subpopulation in the order DOPG>DOPC. At HP = 50%, liposome–protein complexes avoided capture by circulating leukocytes. In this context, monocytes (i.e., CD14+CD3−) represent the subpopulation that maintains the highest grade of uptake.

It is worth noting to observe that massive capture of pre-coated DOTAP liposomes by THP1 cells and leukocyte subpopulations measured in vitro at HP = 5% can be explained by their own biological identity. Indeed, DOTAP–protein complexes are large in size, and their PC is particularly enriched of Immunoglobulins that play a key role in vesicle opsonization[48]. At low plasma concentration, PC of DOTAP is also more enriched of Fetuin that is known to act as an opsonin when it interacts with spermine and other cationic species[49]. At high protein concentration, cellular uptake was generally low and irrespective of the synthetic identity of pristine liposomes. We also observe that small-size DOPC–protein complexes exhibit the lowest particle uptake by immune cells. This result is in full agreement with the observation that most approved liposomal formulations are neutral in charge[8]. In summary, mechanistic investigation of the liposomes' biological identity as a function of protein concentration allowed identifying pre-coated liposomes at HP = 50% as the best candidates for drug-delivery applications in vivo as they are small in size, negatively charged, enriched of anti-opsonins (e.g., Clusterin[28]), and poorly enriched of typical opsonins (e.g., Immunoglobulins, Fetuin etc.).

In an attempt to mimic physiological conditions, next step was exposing both uncoated and pre-coated liposomes to whole blood and evaluating capture by leukocyte subpopulations as a function of time between 0,5 min and 60 min uncoated DOTAP liposomes are avidly internalized by almost all the populations analyzed with cellular uptake in the order: CD19+ > CD14+CD3− > granulocytes > CD56+CD3+ ~CD56+CD3− ~CD3+CD4− > CD3+CD4+ (Fig. 5a). Pre-coating DOTAP liposomes with artificial PC significantly reduced capture by all populations of circulating leukocytes (Fig. 5b) (p-values were evaluated using the Student's t test). On the other side, Fig. 5c–f show that sequestration of uncoated DOPC and DOPG by PBMCs was lower than that of DOTAP and, for each subpopulation in the order DOPG>DOPC. We observe that capture by circulating leukocytes is large by the earliest exposure time (i.e., 0.5 min) and poorly evolves over time. The most significant exception is found for capture of DOPG liposomes by CD14+CD3− leukocytes, where a significant increase in cellular uptake is found between 0.5 min and 5 min incubation.

Numerous in vitro studies have shown that PC of nanoparticle can evolve over time[19,25] with significant impact on biological outcomes (e.g., cellular uptake by cancer cells[25]).

**Temporal evolution of liposomes' biological identity.** To interpret capture by immune cells properly, we explored the temporal evolution of the biological identity of DOTAP liposomes by sodium dodecyl sulfate-polyacrylamide gel electrophoresis (SDS-PAGE). Figure 6a shows the PC of pre-coated DOTAP (labelled as PL in Fig. 6a). When pre-coated DOTAP liposomes are exposed to patient' plasma, their PC is not modified by interaction with plasma proteins. Second, we observe that PCs of uncoated and pre-coated liposomes are established by the earliest exposure time (i.e., 0.5 min), and are stable over time. This finding is in agreement with previous findings by Tenzer et al. showing that PC is rapidly established and does not appreciably change in composition over time[19]. Moreover, Simon et al. have shown that pre-formed coronas remain stable even after nanoparticles are re-introduced to plasma[50]. The results of Fig. 6a also indicate that temporal evolution of leukocytes uptake may be not related to changes in PC composition. In Fig. 6b, we show the protein patterns around uncoated and pre-coated liposomes. This provides a semiquantitative description of the band intensities that are clearly visible in Fig. 6a. The main difference in the protein profiles is found in the molecular weight region between 46 kDa and 56 kDa. In that range MS/MS findings clarify that the most abundant PC components are the γ-chain and the β-chain of fibrinogen (47 kDa and 56 kDa, respectively). Therefore, the results of Fig. 6 let us conclude that liposomes' pre-coating may reduce capture by circulating leukocytes by preventing binding of fibrinogen. This suggestion is in full agreement with previous studies[51,52] showing that fibrinogen is a physiologically relevant ligand for $α_Mβ_2$/Mac 1, the integrin receptor that is critical to leukocyte function and innate immunity in vivo. Languino et al.[53] have demonstrated that binding of fibrinogen to vascular cell receptors induces leukocyte adhesion to endothelium and leukocyte transendothelial migration, which are the earliest events of immune inflammatory responses. More recently, Deng et al.[54,55] showed that fibrinogen bound to some nanoparticles types (e.g., negatively charged gold NPs) undergoes denaturation, activates the integrin receptor Mac-1, and stimulates the NF-κB signaling pathway leading to release of inflammatory cytokines.

As discussed above, among biological processes affecting drug-delivery efficacy and toxicity, the immune recognition of liposomes plays a key role. However, the mechanisms that control the liposome immune recognition and stability are poorly understood. Over the last decades, most in vivo studies have investigated clearance profile and pharmacokinetics of liposomes with different surface potentials[56]. It is well-known that intravenously injected cationic liposomes undergo severe aggregation through electrostatic interactions between cationic lipids and the anionic species in the bloodstream, which results in a reduced circulation times and poor distribution in tumors[57]. Moreover, also the role of IgG on the clearance of cationic particles by macrophages is a consolidated paradigm in drug delivery[58]. It involves binding to Fc receptors, engulfment by lamellipodia, cellular uptake, and subsequent delivery of vesicles to lysosomes for degradation. Of note, all these evidences correlate with the liposomes' biological identity of cationic liposomes (Figs. 1–3). This study also investigated the role of pre-formed PC on the biological identity and the sequestration by immune cells. Of note, pre-coating liposomes with artificial PC made of human plasma proteins avoid capture by THP1 cells and PBMCs in vitro as well as uptake by leukocytes population in patients' whole blood (Figs. 4, 5). According to our findings, we suggest that pre-coating liposomes may allow avoiding sequestration by immune cells thus prolonging blood circulation and the change to reach final destination (i.e., target organs/tissues)[12,48,56,59]. To this end, pre-coating liposomes should satisfy some basic requirements: (i) liposome–protein complexes must be small in size and negatively charged; (ii) content of immunoglobulins in the pre-formed PC must be kept at minimum.

Mechanistic understanding of the interaction between liposome–protein complexes and immune cells let us identify some other implications of our work. Recently, researchers have been attempting to explore the role of immune response in cancer, and macrophages have emerged as a major component of the tumor microenvironment and coordinate various aspects of

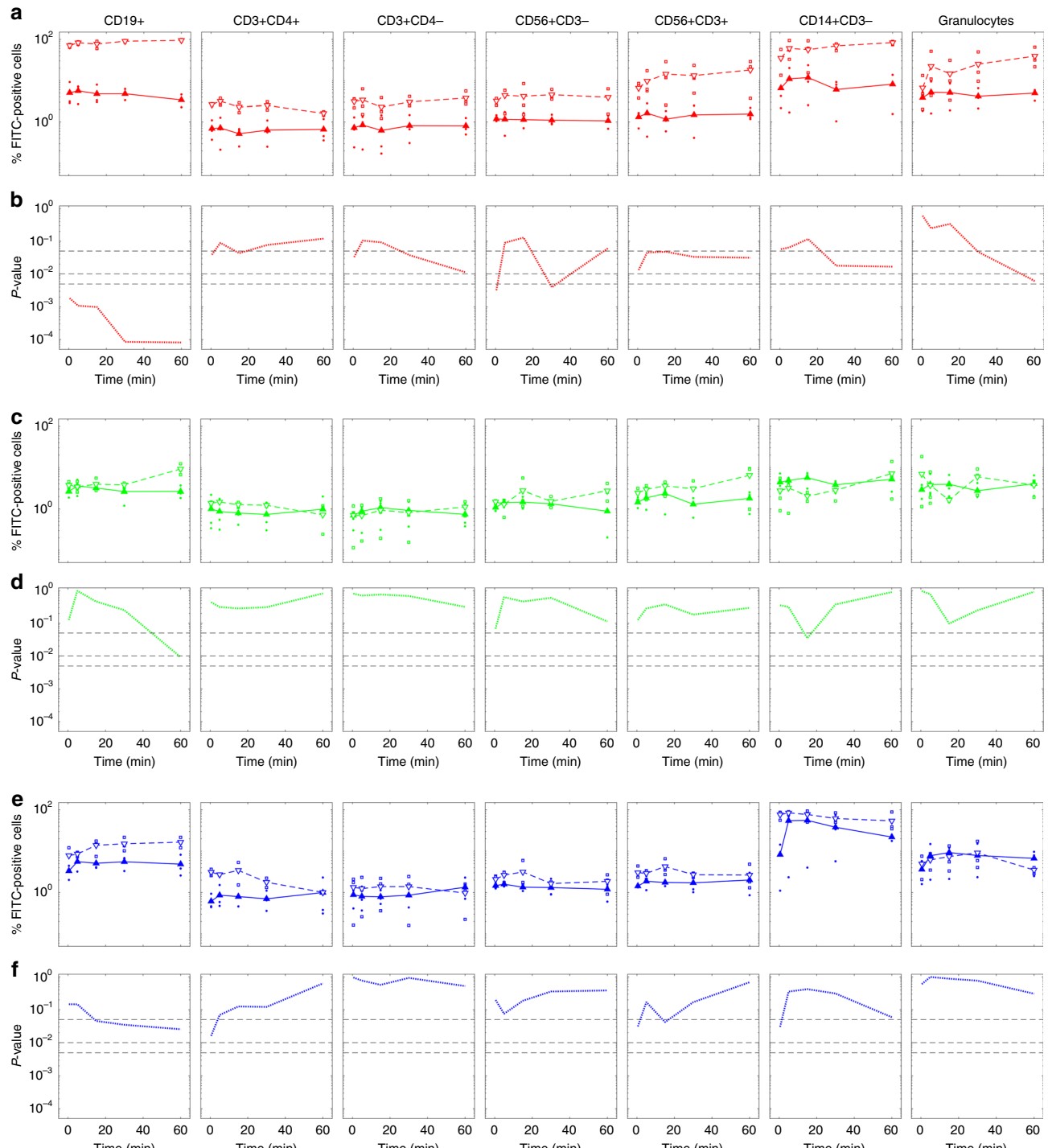

**Fig. 5** Leukocyte uptake of uncoated and pre-coated liposomes in whole blood. Cellular uptake of uncoated (empty triangles) and pre-coated (full triangles) liposomes in whole blood: DOTAP (red, panel **a**), DOPC (green, panel **c**), and DOPG (blue, panel **e**). The fluorescence of internalized liposomes was measured as the percentage of FITC positive cells by gating on distinct leukocyte subpopulations as indicated. The gating strategy was obtained as shown in Supplementary Fig. 3. Each value is the average of duplicate samples ± standard deviation within a single experiment. Statistical significance of difference in cellular uptake was evaluated using the Student's *t* test: DOTAP (red, panel **b**), DOPC (green, panel **d**), and DOPG (blue, panel **f**). From top to bottom dashed lines indicate *p*-values: 0.05; 0.01; 0.005

immunity[60,61]. Preclinical trials have identified tumor-associated macrophages as one of the most promising targets in cancer therapy for the treatment of solid tumors. Our results suggest therefore that cationic liposomes pre-coated with artificial PC at low protein concentration could be ideal nanocarriers for targeting macrophages in cancer. This opens up the great potential to exploit protein corona formation, which will significantly influence the development of novel nanomaterials. As an additional important conclusion, our results suggest that in vitro investigations performed at low plasma concentration may better mimic the behavior of liposomes in vivo thus contributing to predict immune response.

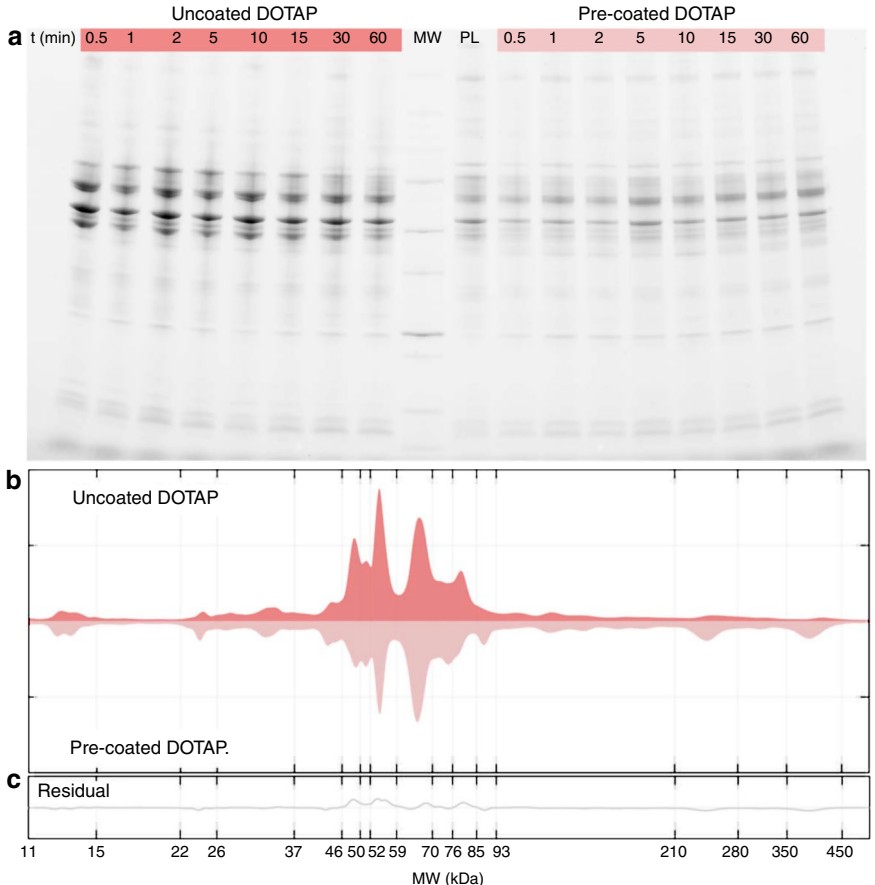

**Fig. 6** Protein corona of uncoated and pre-coated liposomes in patients' plasma. **a** SDS-PAGE gel of human plasma proteins obtained from uncoated and pre-coated DOTAP liposomes following incubation in patients' plasma at different time points. The molecular weights of the proteins in the standard ladder (MW) and the protein pattern of pre-coated liposomes (i.e., before incubation in patients' plasma; labelled as PL) are reported in the middle for reference. **b** One-dimensional (1D) averaged protein profiles of coronas formed around uncoated and pre-coated DOTAP after exposure to patients' plasma. **c** Residual obtained by subtracting the 1D profile of pre-coated from that of uncoated DOTAP

## Methods

**Liposomes preparation**. Cationic DOTAP, DOPC, and DOPG were purchased from Avanti Polar Lipids (Alabaster, AL). DOTAP, DOPC, and DOPG are among the most widely used lipid species for the delivery of drugs and nucleic acids. Each lipid was dissolved in chloroform, and the solvent was evaporated under vacuum for 2 h. Lipid films were hydrated with ultrapure water to a final lipid concentration of 1 mg mL$^{-1}$ and stored at 4 °C. The obtained liposome suspensions of DOTAP and DOPG were extruded 20 times through a 0.1 -μm polycarbonate filter with the Avanti Mini-Extruder (Avanti Polar Lipids, Alabaster, AL). On the other side, DOPC liposomes were sonicated for 20 min.

**Liposome–protein complexes**. Liposome–protein complexes were prepared by incubating liposomes (100 μl) for 1-h at the following HP concentrations: 1%, 2.5%, 5%, 10%, 20%, and 50%. For small angle X-ray scattering (SAXS) and dynamic light scattering (DLS) experiments, samples were characterized without further treatment. For sodium dodecyl sulfate-polyacrylamide gel electrophoresis (SDS-PAGE), bicinchoninic acid assay (BCA assay) mass spectrometry experiments, liposome–protein complexes were isolated by centrifugation for 15 min at 14,000 × *g*. The pellets were washed three times with PBS to remove unbound and loosely bound proteins (the "soft corona") obtaining the so-called "hard corona".

**Ultra-high-performance liquid chromatography electrospray ionization tandem mass spectrometry**. To verify that the largest fraction of liposomes was in the pellet, we performed ultra-high-performance liquid chromatography electrospray ionization tandem mass spectrometry (UHPLC/ESI-MS/MS) experiments. UHPLC/ESI-MS/MS is a powerful method for the microanalysis of many lipid classes in biological samples. UHPLC/ESI-MS/MS system was composed by a triple-stage quadrupole (TSQ) EMR$^{TM}$ (enhanced mass range), coupled to an UHPLC system (UltiMate 3000) (Thermo Fisher Scientific, Bremen, Germany) binary pump via a heated ESI source. The software Xcalibur$^{TM}$ 2.2 (Thermo Fisher Scientific, Bremen, Germany) was used for LC-MS data acquisition and processing. Lipid aliquots of 10 μL were injected via the UHPLC autosampler. The separation

was performed on a Kinetex EVO 100 × 2.1 mm (1.7-μm particle size) (Phenomenex, Torrance, CA, USA). The column was equilibrated at 40 °C. The mobile phase was water (A) and methanol (B), both containing 1 mM ammonium formate and 100 mM formic acid; the flow rate was 300 μL min$^{-1}$. The elution gradient was the following: 0 min (60% B), 0.1−5 min (60−70% B), 5.1−30 (70−99% B), followed by a 5 min washing at 99% phase B and 10 min equilibration at 60% phase B. The total analysis time including column re-equilibration was 45 min aliquots of samples were injected into the system and analyzed at 40 °C. MS Data were acquired in multiple reaction monitoring (MRM) mode, by operating ESI source in positive and negative mode. The tune parameters were set as follows: spray voltage, +3.0 kV in positive mode, 2.5 kV in negative mode; vaporizer temperature, 300 °C in positive mode, 250 °C in negative mode; capillary temperature, 275 °C in positive mode, 320 °C in negative mode; sheath gas pressure, 50 arbitrary units (a.u.) in positive mode, 35 a.u. in negative mode; sweep ion gas pressure 20 a.u. in both positive and negative modes; auxiliary gas pressure 20 a.u. in both positive and negative modes. For each lipid species, three MRM transitions were monitored (Supplementary Fig. 1A). For each lipid, lipid of detection (LOD) and limit of quantification (LOQ) were obtained as follows: (i) DOTAP: LOD = 0.017 ng μL$^{-1}$ and LOQ = 0.014 ng μL$^{-1}$; (ii) DOPC: LOD = 0.015 ng μL$^{-1}$ and LOQ = 0.010 ng μL$^{-1}$; (iii) DOPG: LOD = 0.012 ng μL$^{-1}$ and LOQ = 0.0016 ng μL$^{-1}$. Monthly, the calibration solutions provided by Thermo Fisher Scientific (range m/z 69–2800) were injected in infusion mode for mass calibration and resolution adjustments of the resolving lens and quadrupole. Retention times of lipids were: DOPG ~23.8 min; DOTAP ~25.1 min; DOPC ~26.5 min (Supplementary Fig. 1B). For each lipid, calibration curves were obtained between 0.01 and 3.5 ng μL$^{-1}$ (Supplementary Fig. 1C). When necessary (i.e., when lipid concentration was higher than 3.5 ng μL$^{-1}$), samples were pre-diluted. Dilution factor was properly considered to estimate lipid concentration. Lipid concentrations for each step of the procedure and are reported in Supplementary Fig. 1D. In Fig. S1E, we show the lipid percentage in lipid–protein complexes with respect to bare liposomes (i.e., before exposure to HP). The results are provided as average ± standard deviation of three replicates.

**Transmission electron microscopy**. Liposome–protein complexes (10 μL) were dropped on formvar–carbon-coated copper grids (EMS, PA, USA) and allowed to adsorb for 5 min. The resulting film was stained with a 2% uranyl acetate solution for 1 min at room temperature. Excess of staining solution was adsorbed by the filter paper. Grids were air-dried for 1 h before imaging with TEM Morgagni 268D (Philips, Netherlands). TEM images reported in Supplementary Fig. 2 are in good agreement UHPLC/ESI-MS/MS data of Supplementary Fig. 1.

**Dynamic light scattering and zeta potential**. Size and zeta-potential measurements were carried out using a Zetasizer Nano ZS (Malvern, UK) at 25 °C. To perform size and zeta-potential experiments, 100 μL of liposome–PC complexes were diluted with 900 μL of distilled water (final volume = 1 mL). The results are given as mean ± standard deviation of three replicates.

**Synchrotron small-angle X-ray scattering**. SAXS measurements were performed at the synchrotron light source of Elettra (Trieste, Italy). SAXS patterns were recorded spanning the q-ranges from $0.05$ nm$^{-1}$ to $5$ nm$^{-1}$ with a resolution of $5 \times 10$ nm$^{-3}$ (full width at half maximum). Calibration of detectors was performed with silver behenate powder (d-spacing = 5.838 nm). Typical exposure times were 40 s perrun. The sample was held in a 1 -mm glass capillary (Hilgenberg, Malsfeld, Germany), and the measurements were executed at room temperature. No radiation damage was observed in the SAXS patterns.

**Protein assay**. The amount of plasma proteins bound to liposomes was quantified by using the BCA Assay reagent following the manufacture's protocol (Pierce, Thermo Scientific, Waltham, MA, USA). First liposomes (100 μl) were incubated with HP at the following plasma concentrations: 1%, 2.5%, 5%, 10%, 20%, and 50%. Next, we have collected the liposome–protein complexes using centrifugation for 15 min at $14,000 \times g$, and the pellets were washed three times with PBS. Pellets were resuspended in lysis buffer (10 μL), and each sample was placed in a well of a 96-multi-well plate. In each well, we added 200 μL of BCA Protein Assay reagent. After incubation at 37 °C for 30 min, samples were mixed on a plate shaker. Sample absorbance was measured at 560 nm using a Glomax Discover System (Promega, Madison, WI, USA). The protein concentration was finally calculated using a calibration curve. All results are the average of three independent experiments ± standard deviation.

**1D SDS-PAGE experiments**. Liposome–PC complexes were resuspended in 40 μL of Laemmli Loading buffer ×1 and boiled for 10 min at 100 °C. For DOPC and DOPG sample identical volumes, 20 μL of each sample were loaded on a gradient polyacrylamide gel stain free (4–20% TGX precast gels, BioRad) and run at 100 V for about 150 min. Due to the high protein adsorption ability of DOTAP, samples were diluted (1:10) before loading them on the polyacrylamide gel. Finally, gel images were acquired with a ChemiDoc™ Gel Imaging System (BIO-RAD). Stain-free imaging enables immediate visualization of proteins without gel staining. Images were processed by means of custom MatLab scripts (MathWorks®). Pure HP (i.e., not incubated with liposomes) was subjected to the same centrifugation procedure illustrated above to exclude that liposome–protein pellets were biased by unbound proteins.

**Tandem mass spectrometry**. To perform nano liquid chromatography tandem mass spectrometry (nanoLC-MS/MS) experiments, we hydrated lipid films with a dissolving buffer (Tris-HCl, pH 7.4, 10 mmol L$^{-1}$; NaCl, 150 mmol L$^{-1}$; EDTA, 1 mmol L$^{-1}$). Obtained liposomes were dimensioned by extrusion. To this end, liposomal dispersions were extruded slowly 20 times through a 100 -nm polycarbonate carbonate filter by means of the Avanti Mini-Extruder (Avanti Polar Lipids, Alabaster, AL). Unilamellar liposomes were stored at 4 °C until use. Liposome–protein complexes were made exposing liposomes to HP at 37 °C. After 1-h incubation, samples were centrifuged 15 min at $14,000 \times g$ followed by pellet washing with dissolving buffer. Washing procedure was repeated three times to eliminate the soft corona. Next, the pellet was resuspended in 40 μl of 8 mol L$^{-1}$ urea, and 50 mmol L$^{-1}$ NH$_4$HCO$_3$ (pH = 7.8). Afterwards, protein solution was treated and prepared for the liquid chromatography mass spectroscopy through a procedure described in our recent publication[62] in details.

**Cell culture**. Human monocyte cell line THP1 was purchased from ATCC (ATCC® TIB-202™), and were maintained in the RPMI-1640 medium supplemented with 10% fetal bovine serum until use.

**Ethic statement**. Informed and written consent in accordance with the Declaration of Helsinki was obtained from all healthy donors, and approval was obtained from the Ethics Committee of the Sapienza University of Rome.

**Flow cytometry**. To investigate cellular uptake of nanoparticles in THP1 cell line, each of the three liposomal formulations was synthesized with addition of DOPE-NBD (fluorescent lipid/total lipid = 5/1000 mol/mol). Bare liposomes and liposome–protein complexes were administered to cells with serum-free medium. THP1 cells were plated at 500.000 cells mL$^{-1}$ in 12-well dishes, and then were incubated for 3 h with 10 μg mL$^{-1}$ of NBD-labeled liposomes in the Optimem medium. After the treatment, cells were washed with cold PBS and then run on a BD LSFORTESSA (BD Biosciences, San Jose, CA, USA). Cells were gated using forward versus side scatter to exclude debris. The data were analyzed using FlowJo software (FlowJo LLC data analysis software, Ashland, OR, USA) as elsewhere described[63].

**Particle sequestration from circulating leukocytes**. PBMCs were isolated from peripheral blood of healthy donors by Ficoll-Hypaque gradient centrifugation. Cells were plated at $1 \times 10^6$ cells mL$^{-1}$, and then were incubated for 1 h at 37 °C with 10 μg/mL of NBD-labeled liposomes in the RPMI medium. After the treatment, cells were washed with PBS and then labeled with the following diluted antibodies: anti-CD3/BV510 (cat. 564713, dilution 1:50), CD56/BV421 (cat.562751, dilution 1:50), anti-CD4/APC (cat.555349, dilution 1:10), anti-CD14/PerCP (cat. 340585, dilution 1:50), anti-CD45/allophycocyanin-H7 (cat. 560178, dilution 1:50), and anti-CD19/PE-Cy7 (cat.557835, dilution 1:100), all from BD Bioscience. In some experiments, 120 μl of peripheral whole blood was incubated with 50 μg ml$^{-1}$ of NBD-labeled liposomes for different times (30 s or 1, 5, 15, 30, and 60 min) at 37 °C and washed by centrifugation using 2 ml of physiological solution. Red blood cells were then lysed with a buffer containing 155 mM NH$_4$Cl, 12 mM NaHCO$_3$, 0.1 mM EDTA. After washing by centrifugation, cells were labeled with a mixture of fluorescent antibodies as described above. The fluorescence of internalized liposomes was evaluated by immunofluorescence and FACS analysis using a FACSCanto (BD Biosciences, San Jose, CA) and measured as the percentage of FITC positive cells by gating on distinct leukocyte subpopulations (Supplementary Fig. 3). The data analysis was performed using the FlowJo program[63].

**Statistical information**. The results are reported as mean ± standard deviation, and are represented as error bars in the graphs. The number of replicates for each sample varied among experiments, as follows: $n = 3$ for size, zeta-potential measurements, and proteomics experiments, $n = 2$ for 1D SDS-PAGE and FACS analysis on the THP1 cell line. For PBMCs experiments, $n = 3$ for all the investigated samples, except to DOTAP at HP = 5%, for which $n = 1$. To interpret and discuss the experimental data neither Student's $t$ test nor ANOVA test were necessary.

**Reporting summary**. Further information on research design is available in the Nature Research Reporting Summary linked to this article.

## Data availability
The datasets generated during and/or analyzed during the current study are available from the corresponding author on reasonable request.

## Code availability
Custom codes used for SDS-PAGE are available from the corresponding author on request.

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

## Acknowledgements

S.P. is recipient of a fellowship (ID 19319) granted from AIRC foundation for cancer research. M.M. would like to thank funding from Precision Health Program at Michigan State University.

## Author contributions

Conceptualization, A.L., D.P., M.M., I.S., A.Z. and G.C.; methodology, A.L., D.P., H.A., M.M., I.S., A.Z. and G.C.; software, L.D. and R.Z.C.; validation, F.G., S.P., E.V. and A.L.C.; formal analysis, L.D., E.V. and R.Z.C.; investigation, F.G., S.P., L.D., E.V., A.L.C., and L.M.; resources, H.A., A.L., L.M., A.Z. and G.C.; data curation, L.D.; writing—original draft preparation, F.G. and G.C.; writing—review and editing, D.P., M.M., I.S., A.Z. and G.C.; visualization, S.P. and L.D.; supervision, A.L., D.P., I.S., A.Z. and G.C.; project administration, G.C.; funding acquisition, H.A., I.S., A.Z., M.M., and G.C.

## Additional information

**Competing interests:** The authors declare no competing interests.

