## [Peer Review File · Nature Communications]

Reviewers' comments:

Reviewer #1 (Remarks to the Author):

The manuscript titled "Biological identity of liposomes controls sequestration by immune cells in vivo" by Giulimondi et al aims to identify the link between biological identity of liposomes formed under in vitro incubation on their sequestration from peripheral blood mononuclear cells. The authors claim that this has a crucial role for clinical translation of liposomal drug delivery.

There several major issues with this work as explained below:

1. The design of the study completely contradicts the purpose. The authors tested the incubation of liposomes of different lipid composition with increasing concentrations of proteins in order to identify the composition and the effect of early protein adsorption when the particles first introduced into the circulation. However, they designed that in an in vitro setup over 1h incubation under static conditions, which completely defeated the purpose.
2. The separation method of liposomes after protein corona formation from the plasma protein is not convincing at all. It is well known that liposomes do not completely precipitate after centrifugation at 14000 rpm, which indicate that the authors excluding some fraction of liposomes in their analysis. This will definitely impact on the reliability of their data. Did the author check that the supernatant after washing does not have liposomes???
3. The results presented are not novel. Many studies have been published recently on protein corona formation on liposomes under in vivo condition and even after recovery from cancer patients treated with liposomal anticancer drugs. The authors did not mention or refer to these studies at all in their manuscript.
4. The article generally is not well written and there are several mistakes e.g. Figure S1 the number of the lanes on the gel electrophoresis for one of the liposomes systems is missing. No indication of the molecular weight of the bands separated on the gel. The lower protein concentration used was stated to be 2.5% and 1% , not sure which one is the correct.

Reviewer #2 (Remarks to the Author):

The authors did a systemic in vitro study to understand the protein corona (PC) effects on the liposomes with three different surface charges. For my understanding, the authors mainly focused on the effects of plasma protein concentration on the PC bound to liposomes, which will further influence the cellular uptake of liposomes. I must say it is a good point of view. However, there are several flaws in the study.

Figure 3 showed the composition of PC on the liposomes. I am confused about the authors' interpretation of the results. The table S2 showed the protein bound to liposome increased for all the liposomes. However, in the line of 167 and 168, the authors only made conclusions for DOPC and DOPG. What about DOTAP? It seems to me that the PC composition of DOTAP also remains largely unchanged (Fig. 3D), whereas the PC composition of DOPC and DOPG changed a lot according to the figure 3E and 3F. I also would like to see the statistical analysis of the figure 3D-F.

The uptake of liposomes by THP-1 cells and PBMCs showed that the DOTAP has the most uptake compared to DOPC and DOPG. Despite the IgG binding to the liposomes, another reason may be the positive charge of DOTAP which could interact with the negative charged cell membrane. Even though the DOTAP after PC binding showed negative charged, the PC formation is a dynamic process which was shown by Simberg lab published in Nature Nanotechnology. The authors did not incubate liposomes only without PC with the cells to see if the uptake could happen too. Actually, Simberg lab has reported that the complement proteins play important role in the recognition of nanoparticles by immune cells in the blood. Perhaps the authors can use Simberg lab's methods to directly incubate the

liposomes with whole human blood and try to analyze the leukocyte types that internalize the liposomes. For the leukocyte uptake of nanoparticles, I believe the uptake study conducted in the whole blood, rather than using plated cells which are not the case in vivo.

Another note to the authors is that the PC formation and composition for different individuals also varies. That would explain the in vitro data does not match in vivo data. It is good to develop a few in vitro tests to evaluate nanoparticle interactions with plasma protein. However, eventually the best test for nanoparticle use in clinics would be specific to each individual using his or her own blood.

In sum, I would suggest the authors revising the manuscript.

Reviewer #3 (Remarks to the Author):

The manuscript describes the formation of protein corona in different ratios of blood plasma and liposomes with different surface charge. This is described as a model to study the development of protein corona in blood. It describes the amount of proteins bound, the nature of the proteins, and the changes in surface chemistry (Z-potential) and size and morphological changes of the liposomes after protein binding. It further links these changes with differences in cellular uptake.

The authors claim that the approach to look at different protein /surface ratio is novel. This may be true specifically for nano-sized liposomes. However, there is a large body of literature describing these situations on many other nanoparticles of different materials. Important literature is not taken into consideration. One such area is what drives the protein dependent aggregation which could explain many of the observed data (see also further below). Example of literature in this field: Nano Research 8, 2733; ACS Nano, 6, 8962;

Due to the possible protein dependent aggregation mechanisms, following the progress in blood by step wise adding more plasma to naked liposomes are probably not the best way to conduct the experiment. At lower protein concentrations larger aggregates may be formed by multiple protein interactions. These aggregates are not formed in high protein concentrations. Due to the multiple interactions in the aggregates formed at low protein concentration is not clear, or even likely that the aggregates would disintegrate if transferred into high protein concentrations. The authors themselves indicate that in the discussion by stating "Recent studies have shown that the pre-formed corona remains stable...." (Reference is missing). And by stating "... in vitro results at high plasma concentration do not match most clearance results..." which to me indicates that the complexes are not the same.

The manuscript could be improved by data where the plasma concentration is increased using the same liposomes from concentration to concentration. Possibly the formed aggregates in low concentrations will be stable also in high concentrations.

Technically using only centrifugation at fairly low speed is risky as only high density aggregates/particles may be collected. The authors should use additional separation techniques and/or make sure that a vast majority of the lipids/liposomes are in the pellet.

In its present form the article may raise a discussion in the field. It is for the field a well carried out study with all the normally required characterizations, adding new knowledge to the field. However, the approach to step wise add plasma to naked particles is common, and does probably not correctly describe the development of protein corona / aggregation state in blood after injection. Together with the standard approach of separating/isolating protein/liposome complexes the article is rather confirming what is known and believed in the protein corona field.

I do not recommend publication of the manuscript in Nature communication in its present state.

However, the original questions are very important and interesting and the manuscript has the potential to be improved but a new evaluation is in that case needed.

Regards

Tommy Cedervall

Reviewer #1

1. Reviewer's comment: The design of the study completely contradicts the purpose. The authors tested the incubation of liposomes of different lipid composition with increasing concentrations of proteins in order to identify the composition and the effect of early protein adsorption when the particles first introduced into the circulation. However, they designed that in an in vitro setup over 1h incubation under static conditions, which completely defeated the purpose.

1. Response by the authors: In the revised version of the manuscript testing the incubation of liposomes with increasing concentrations of proteins was aimed at obtaining pre-coated liposomes with optimized biological identity (please see also responses # 2 given to Reviewer 3). To this end, 1-hour static incubation is line with this purpose.

2. Reviewer's comment: The separation method of liposomes after protein corona formation from the plasma protein is not convincing at all. It is well known that liposomes do not completely precipitate after centrifugation at 14000 rpm, which indicate that the authors excluding some fraction of liposomes in their analysis. This will definitely impact on the reliability of their data. Did the author check that the supernatant after washing does not have liposomes???

2. Response by the authors: Liposomes do not completely precipitate at 14000 rpm, the Reviewer correctly states. On the other hand, liposome-protein complexes massively precipitate. It is well known that, even at the lowest plasma concentration, the amount of protein is enough to decorate all the liposomes. However, to provide quantitative demonstration of this, here we performed dynamic light scattering (DLS) experiments. As the Reviewer knows, in DLS the scattering intensity it is measured in photons per second, or most often as in the Malvern Zetasizer (Malvern, UK) in kilo counts per second, kcps for short. The Zetasizer can detect the scattering from very low concentration as well as from very high concentration samples. To compare the signal strength from different samples, a useful parameter is the derived count rate (DCR) that is proportional to sample concentration, size or both. For a given sample, the higher derived count rate usually indicates higher concentration. As Figure clearly shows, DCR of the starting solution at HP=50% is larger than that of complexes at HP=5%. As samples contain the same number of liposomes and the size of DOTAP-protein complexes at HP=50% is smaller than that of complexes at 5% this result indicates that the extra-scattering DCR for complexes prepared at HP=50% arises from excess scattering objects, i.e. unbound proteins. After centrifugation at 14000 rpm, the DCR from supernatants ("supernatant 1") is much lower than that of the starting solution, thus ensuring that most liposomes are in the pellet (please also consider that DCR in "supernatant 1" may arise from unbound proteins). Then, the pellet is washed three times with PBS. After each washing step, the supernatant was examined by DLS. As Figure clearly shows, DCR decreases monotonously by two order of magnitude thus assuring that the supernatant does not contain any excess material and the pellet only contains tightly bound proteins.

Figure S1. Separation of liposome-protein complexes. The "derived count rate" (DCR) measured in kilo counts per second (kcps) is a calculated parameter in the Zetasizer Nano software (Malvern, UK) and is representative of the scattering intensity that would be measured in the absence of the laser attenuation filter. As such, DCR is proportional to sample concentration, size of particles or both. DCR of liposome-protein complexes at HP=5% and 50% before centrifugation (starting solution) is used as a reference. Liposome-protein complexes were isolated by centrifugation for 15 min at 14,000 rpm. After centrifugation at 14000 rpm, the DCR from supernatants ("supernatant 1") is much lower than that of the starting solution, thus ensuring that most liposomes are in the pellet. The pellet was washed three times with PBS to remove unbound and loosely bound proteins (the 'soft corona') obtaining the so-called 'hard corona'. After each washing step, the supernatant was collected and used for DLS analysis. As the evolution of the DCR shows, the washing procedure reduced the DCR by more than two orders of magnitude. This finding unambiguously indicates that the largest fraction of liposome-protein complexes was in the pellet and that washing eliminates loosely bound proteins.

3. Reviewer's comment: The results presented are not novel. Many studies have been published recently on protein corona formation on liposomes under *in vivo* condition and even after recovery from cancer patients treated with liposomal anticancer drugs. The authors did not mention or refer to these studies at all in their manuscript.

3. Response by the authors: Our study is completely original as it is a mechanistic investigation of the relationship between the biological identity of liposomes and the capture by immune cells *in vivo*. As such, the purpose of the work is totally different from clarifying the PC of liposomal anticancer drugs *in vivo*. However, to increase the significance of the paper, we also cited the following papers:

- Hadjidemetriou, M., Al-Ahmady, Z. & Kostarelou, K. Time-evolution of *in vivo* protein corona onto blood-circulating PEGylated liposomal doxorubicin (DOXIL) nanoparticles. *Nanoscale* 8, 6948-6957 (2016).
- Hadjidemetriou, M. et al. The Human *In Vivo* Biomolecule Corona onto PEGylated Liposomes: A Proof-of-Concept Clinical Study. *Advanced Materials*, 1803335 (2018).
- Hadjidemetriou, M., Al-Ahmady, Z., Buggio, M., Swift, J. & Kostarelou, K. A novel scavenging tool for cancer biomarker discovery based on the blood-circulating nanoparticle protein corona. *Biomaterials* 188, 118-129 (2019).

4. Reviewer's comment: The article generally is not well written and there are several mistakes e.g. Figure S1 the number of the lanes on the gel electrophoresis for one of the

liposomes systems is missing. No indication of the molecular weight of the bands separated on the gel. The lower protein concentration used was stated to be 2.5% and 1% , not sure which one is the correct.

4. Response by the authors: Reviewer is gratefully acknowledged for her/his careful reading of the manuscript. All mistakes and typos have been corrected. The manuscript has been also revised by a native speaker.

Reviewer #2

1. Reviewer's comment: Figure 3 showed the composition of PC on the liposomes. I am confused about the authors' interpretation of the results. The table S2 showed the protein bound to liposome increased for all the liposomes. However, in the line of 167 and 168, the authors only made conclusions for DOPC and DOPG. What about DOTAP? It seems to me that the PC composition of DOTAP also remains largely unchanged (Fig. 3D), whereas the PC composition of DOPC and DOPG changed a lot according to the figure 3E and 3F. I also would like to see the statistical analysis of the figure 3D-F.

1. Response by the authors: Figure 3 and related data (Fig. S1 in the former version of the manuscript) have been eliminated from the revised version of the manuscript. To increase the readability of the manuscript, the authors compared composition of the liposome coronas by nano-liquid chromatography MS/MS.

2. Reviewer's comment: The uptake of liposomes by THP-1 cells and PBMCs showed that the DOTAP has the most uptake compared to DOPC and DOPG. Despite the IgG binding to the liposomes, another reason may be the positive charge of DOTAP which could interact with the negative charged cell membrane. Even though the DOTAP after PC binding showed negative charged, the PC formation is a dynamic process which was shown by Simberg lab published in Nature Nanotechnology. The authors did not incubate liposomes only without PC with the cells to see if the uptake could happen too. Actually, Simberg lab has reported that the complement proteins play important role in the recognition of nanoparticles by immune cells in the blood.

2. Response by the authors: Uptake of uncoated liposomes (i.e. without PC) by THP1 has been reported in Fig. 4A (HP=0%). Uncoated DOTAP is more internalized than uncoated DOPC and DOPG. As Reviewer # 2 correctly states, this is likely due to the positive charge of DOTAP that promotes electrostatic interaction with negatively charged surface proteoglycans of THP1 cells. At low protein concentration (HP< 5 %), DOTAP-protein complexes are still positive in charge, thus their uptake could be partly influenced by not-specific electrostatic interaction with cells. However, at HP=5% DOTAP-protein complexes are neutrally charged so no electrostatic interaction with THP1 cells occurs. In this case, only size-dependent mechanism may account for the observed increase in cellular uptake.

To address the issue raised by the Reviewer the following sentence has been added (**Line 175**): *"In the absence of PC, cellular uptake of DOTAP was much higher than that of DOPC and DOPG. This is likely due to its cationic charge that promotes not specific cellular association by electrostatic attraction with negatively charged plasma membrane."*

The Nature Nanotechnology paper has been introduced in the reference list of the revised version of the paper.

3. Reviewer's comment: Perhaps the authors can use Simberg lab's methods to directly incubate the liposomes with whole human blood and try to analyze the leukocyte types that internalize the liposomes. For the leukocyte uptake of nanoparticles, I believe the uptake study conducted in the whole blood, rather than using plated cells which are not the case in vivo.

3. Response by the authors: Reviewer #2 is gratefully acknowledged for her/his comment that allowed the authors to increase the significance of the work. To address this issue, both uncoated and pre-coated liposomes were incubated in whole blood from healthy donors. Cellular uptake by leukocytes populations was evaluated after incubation in whole blood at several time points (30 sec or 1, 5, 15, 30 and 60 min). New results have been reported in Fig. 5 of the revised version of the manuscript (it is reported below).

Figure 5. Leukocyte uptake of uncoated and pre-coated liposomes in whole blood. Cellular uptake of uncoated (empty triangles) and pre-coated (full triangles) liposomes in whole blood: DOTAP (red, panel A), DOPC (green, panel C) and DOPG (blue, panel E). The fluorescence of internalized liposomes was measured as the percentage of FITC positive cells by gating on distinct leukocyte subpopulations as indicated. Statistical significance of difference in cellular uptake was evaluated using the Student's t-test: DOTAP (red, panel B), DOPC (green, panel D) and DOPG (blue, panel F). From top to bottom dashed lines indicate p-values: 0,05; 0,01; 0,005.

4. Reviewer's comment: Another note to the authors is that the PC formation and composition for different individuals also varies. That would explain the in vitro data does not match in vivo data. It is good to develop a few in vitro tests to evaluate nanoparticle interactions with plasma protein. However, eventually the best test for nanoparticle use in clinics would be specific to each individual using his or her own blood.

4. Response by the authors: Reviewer 2 is right. However, pre-coating liposomes with human plasma properties was aimed at creating an artificial PC with optimized properties for drug delivery that stays stable after introduction in the patient's bloodstream. To account for inter-individual variability, we incubated liposomes in whole blood from distinct individuals. As our results show (Fig. 5), pre-coating dramatically reduces cellular uptake by leukocytes populations and this effect is general, i.e. it is poorly dependent on inter-individual variability.

Reviewer #3

1. Reviewer's comment: The authors claims that the approach to look at different protein /surface ratio is novel. This may be true specifically for nano-sized liposomes. However, there is a large body of litterature describing these situations on many other nanoparticles of different materials. Important litterature is not taken in concideration. One such area is what drives the protein dependent aggregation which could explain many of the observed data (see also further below). Example of litterature in this field: Nano Research 8, 2733; ACS Nano, 6, 8962;

1. Response by the authors: Reviewer 3 is gratefully acknowledged for his comment. These works were commented on and inserted in the reference list of the work.

(Line 108): All these results are in good agreement with previous results obtained using other nanoparticles of different materials. In particular, Cukalevski et al. showed that low protein concentrations pro-moted formation of large clusters of polystyrene and polymethyl methacrylate nanoparticles, whereas high protein concentrations induced small-size aggregates ⁴¹.

(Line 251): "More recently, Deng et al.^{56, 57} showed that fibrinogen bound to some nanoparticles types (e.g. negatively charged gold NPs) undergoes denaturation, activates the integrin receptor Mac-1 and stimulates the NF- κ B signalling pathway leading to release of inflammatory cytokines."

2. Reviewer's comment: Due to the possible protein dependent aggregation mechanisms, following the progress in blood by step wise adding more plasma to naked liposomes are probalby not the best way to conduct the experiment. At lower protein concentrations larger aggregates may be formed by multiple protein interactions. These aggregates are not formed in high protein concentrations. Due to the multiple interactions in the aggregates formed at low protein concentrationt is not clear, or even likely that the aggregates would disintergrate if transferred into high protein concentrations. The author themselves indicate that in the discussion by stating " Recent studies have shown that the pre-formed corona remains stable..." (Reference is missing). And by stating "... in vitro results at high plasma concentration do not match most clearance results..." which to me indicates that the complexes are not the same. The manuscript could be improved by data where the plasma concentration is increased using the same liposomes from

concentration to concentration. Possibly the formed aggregates in low concentrations by will be stable also in high concentrations.

2. Response by the authors: the comment of Reviewer 3 was highly appreciated as it allowed the authors to focus a central point of the work. The question was whether the biological identity of liposomes at 50% of human plasma is the same as that obtained through a two-step procedure consisting of a first incubation at low plasma concentrations (HP=5%) for 1-hour at 37 °C followed by addition of human plasma to the same sample to reach the final plasma concentration of 50%. To follow Reviewer's indication, we increased plasma concentration using the same liposomes from concentration to concentration. In detail, liposomes were incubated for 1-hour at HP=5%, then plasma concentration was increased using the same sample and brought to HP=50%. As Reviewer 3 correctly predicted, aggregates formed at low concentration were not stable. They underwent disaggregation and, within experimental uncertainties, were equal to those formed under a single incubation at HP=50%. On the other side, liposome-protein complexes prepared at HP=50% were found to be stable even if diluted.

These experiments unambiguously showed that pre-coated liposomes are stable if they are pre-incubated in 50% plasma. This stability upon further interaction with plasma proteins makes them excellent candidates for drug delivery and makes it possible to explore the relationship between biological identity and physiological response. In summary, the Reviewer's suggestion allowed us to better focus the implications of our work as well as to extensively rewrite it. In the revised version of the manuscript, the characterization of liposome-protein complexes as a function of plasma concentration was aimed at identifying complexes whose biological identity is optimal for drug delivery in terms of size, surface charge and corona composition and that is stable upon exposure to a protein-enriched environment.

3. Reviewer's comment: Technically using only centrifugation at fairly low speed is risky as only high density aggregates/particles may be collected. The authors should use additional separation techniques and/or make sure that a vast majority of the lipids/liposomes are in the pellet.

3. Response by the authors: Reviewer's comment was highly appreciated. This was checked by the authors (please see response # 2 given to Reviewer 1).

Additional changes

To interpret capture by immune cells properly, we applied SDS PAGE to explore the temporal evolution of the protein corona of uncoated and pre-coated cationic liposomes in the plasma of the healthy volunteers. New results are reported in Fig. 6 provided a mechanistic explanation of the stealth effect provided by the artificial corona.

Figure 6. Protein corona of uncoated and pre-coated liposomes in patients' plasma. (A) SDS-PAGE gel of human plasma proteins obtained from uncoated and pre-coated DOTAP liposomes following incubation in patients' plasma at different time points. The molecular weights of the proteins in the standard ladder (MW) and the protein pattern of pre-coated liposomes (i.e. before incubation in patients' plasma; labelled as "PL") are reported in the middle for reference. **(B)** One-dimensional (1D) averaged protein profiles of coronas formed around uncoated and pre-coated DOTAP after exposure to patients' plasma. **(C)** Residual obtained by subtracting the 1D profile of pre-coated from that of uncoated DOTAP.

The following text was added (**Line 232**): "*Fig. 6A shows the PC of pre-coated DOTAP (labelled as "PL" in Fig. 6A). When pre-coated DOTAP liposomes are exposed to patient' plasma, their PC is not modified by interaction with plasma proteins. Second, we observe*

that PCs of uncoated and pre-coated liposomes are established by the earliest exposure time (i.e. 0,5 min) and are stable over time. This finding is in agreement with previous findings by Tenzer et al. showing that PC is rapidly established and does not appreciably change in composition over time²¹. Moreover, Simon et al. have shown that pre-formed coronas remain stable even after nanoparticles are re-introduced to plasma⁵². Results of Fig. 6A also indicate that temporal evolution of leukocytes uptake may be not related to changes in PC composition. In Fig. 6B we show the protein patterns around uncoated and pre-coated liposomes. This provides a semiquantitative description of the band intensities that are clearly visible in Figure 6A. The main difference in the protein profiles is found in the molecular weight region between 46 kDa and 56 kDa. In that range MS/MS findings clarify that the most abundant PC components are the α -chain and the β -chain of fibrinogen (47 kDa and 56 kDa respectively). Therefore, results of Fig. 6 let us conclude that liposomes' pre-coating may reduce capture by circulating leukocytes by preventing binding of fibrinogen. This suggestion is in full agreement with previous studies^{53, 54} showing that fibrinogen is a physiologically relevant ligand for $\alpha_M\beta_2$ /Mac 1, the integrin receptor that is critical to leukocyte function and innate immunity in vivo. Languino et al.⁵⁵ have demonstrated that binding of fibrinogen to vascular cell receptors induces leukocyte adhesion to endothelium and leukocyte transendothelial migration, which are the earliest events of immune inflammatory responses. More recently, Deng et al.^{56, 57} showed that fibrinogen bound to some nanoparticles types (e.g. negatively charged gold NPs) undergoes denaturation, activates the integrin receptor Mac-1 and stimulates the NF- κ B signalling pathway leading to release of inflammatory cytokines."

Reviewers' comments:

Reviewer #1 (Remarks to the Author):

Reviewer 1 reply:

We would like to thank the authors to revise the manuscript taking into account our recommendations.

We are happy with the edits made apart from point 2 as we believe that the actual account of the liposomes in the supernatant should be quantified using either Stewart's assay or radiolabelled liposomes to prove the efficiency of the separation. The data should also be backed up with TEM imaging to prove that no liposomes are present in the supernatant.

Reviewer #2 (Remarks to the Author):

The authors well addressed the reviewers' comments and added enough data to complete the story with a focused angle of view. I suggest to publish the manuscript in Nature Communications.

Reviewer #3 (Remarks to the Author):

In the first round of reviews there were sever concerns about the novelty of the manuscript and several technical questions. I do not think that either of the aspects have been well answered. Especially as the author claim that the aim has changed. However, the claim that a preformed corona on nanoparticles change the biological fate is not novel.

One of the technical questions was that it is not clear that all lioposomes have been pelleted after incubation with blood plasma. The authors try to measure the the pelleting by determining the DLS signal after centrifugation. This is not a good method to use for several reasons. I give some examples. In most DLS instrument he auto attenuation function regulates the strength of the laser and the sensitivity of the detectors. This means that the setting is optimized for each reading and the concentration of the measured particles can not be compared from sample to sample. If the laser and detector is locked it might be possible but the authors, as far as I have seen has not reported the settings. And also then for a specific setting larger particles would scatter more light and at that setting small particles may not be detected. As the authors themselves point out larger aggregates of liposomes and proteins pellets easily. Small aggregates, maybe with a single liposome would be much more difficult to detect with DLS especially in a high protein background.

Althoughthe manuscript has interesting result the lack novelty and technical shortage (very common in the field) makes it unsuitable for ublication in Nature Communications

Reviewer #1

1. Reviewer's comment: We would like to thank the authors to revise the manuscript taking into account our recommendations. We are happy with the edits made apart from point 2 as we believe that the actual account of the liposomes in the supernatant should be quantified using either Stewart's assay or radiolabelled liposomes to prove the efficiency of the separation. The data should also be backed up with TEM imaging to prove that no liposomes are present in the supernatant.

1. Response by the authors: Reviewer 1 is gratefully acknowledged for her/his comments on the revised version of the paper. To prove the efficiency of separation we used ultra-high-performance liquid chromatography electrospray ionization tandem mass spectrometry (UHPLC/ESI-MS/MS). UHPLC/ESI-MS/MS is a very powerful method for estimating very low concentration of lipids in biological samples (see references 65-67 of the revised version of the manuscript). After centrifugation, we quantified lipid concentration in the supernatant for each of the six experimental conditions (i.e. lipid species and HP percentage). Lipid concentration in the supernatant ranged between 26.5 ng/ μ L (DOTAP liposomes incubated in 50% HP) and 42.8 ng/ μ L (DOPC liposomes incubated in 5% HP). Considering the initial lipid concentration (1 mg/ml), this means that the percentage of lipid in the supernatant spanned between 2.6% and 4.3%. Then, the pellets were washed three times with PBS to remove unbound and loosely bound proteins (the 'soft corona') obtaining liposomes coated by the so-called 'hard corona'. To check whether washing steps reduced lipid amount in lipid-protein complexes, we quantified liposome concentration in the washing buffer after each of the three washing steps. This procedure allowed us to accurately estimate the percentage of lipid in lipid-protein complexes (i.e. in the pellet) with respect to bare liposomes (i.e. before exposure to HP). This percentage ranged between 95.6% (DOPC exposed to 5% HP) and 97.3% (DOTAP exposed to 50% HP).

Figure S1. Multiple reaction monitoring (MRM) transitions (A) and chromatograms (B) of DOTAP, DOPC and DOPG. (C) Calibration curves for DOTAP (red points), DOPC (green points) and DOPG (blue points) in the concentration range between 0,01 ng/ μ L and 3,5 ng/ μ L. Solid lines represent the best linear fits to the data. R² values were: 0,9985 (DOTAP), 0,9995 (DOPC) and 0,9991 (DOPG). (D) Liposome-protein complexes were isolated by centrifugation for 15 min at 14,000 rpm. Then, pellets were washed three times with PBS to remove unbound and loosely bound proteins (the 'soft corona') obtaining the so-called 'hard corona'. Lipid concentration was calculated in the supernatant (i.e. after centrifugation) as well as after each of the three washing steps. When lipid concentration was out of the linear range (i.e. higher than 3,5 ng/ μ L), samples were pre-diluted. Dilution factors were considered to estimate lipid concentration. (E) Lipid percentage in lipid-protein complexes with respect to bare liposomes (i.e. before exposure to HP). Results are provided as average \pm standard deviation of three replicates.

According to the Reviewer's suggestions, we also performed TEM experiments (Figure S2). As a control, first we performed TEM experiments on liposome-protein complexes before centrifugation (Fig. S2A). Then, liposome-protein complexes were isolated by centrifugation for 15 min at 14,000 rpm. In most cases, TEM images of the supernatant did not contain any trace of vesicles (not reported). When vesicle-like structures were seen, they were prevalently bare liposomes (Fig. S2B). This suggests that the minor fraction of lipids found in the supernatant (as determined by UHPLC/ESI-MS/MS), was due to uncoated liposomes (i.e. in the absence of biomolecular corona). Then, pellets were washed three times with PBS. Finally, TEM images were acquired on the washing buffer after each of the three washing steps (Fig. S2, panels C-E). As evident, no plain evidence of liposome-protein complexes was found. We only detected a weak signal likely due to free proteins.

Figure S2. (A) Representative transmission electron microscopy (TEM) image of DOTAP-protein complexes after 1h incubation with 50% human plasma before centrifugation. Then, liposome-protein complexes were isolated by centrifugation for 15 min at 14,000 rpm. In most cases, TEM images of the supernatant did not contain any trace of vesicles (not reported). When vesicle-like structures were seen (B), they were prevalently bare liposomes). This suggests that the minor fraction of lipids found in the supernatant (as determined by UHPLC/ESI-MS/MS), was due to uncoated liposomes (i.e. in the absence of biomolecular corona). Then, pellets were washed three times with PBS. TEM images were acquired on the washing buffer after each of the three washing steps (C-E). As evident, no plain evidence of liposome-protein complexes was found. We only detected a weak signal likely due to free proteins. Figure legend: scale bars correspond to 200nm. Scale bars in the insets correspond to 100nm.

1. Reviewer's comment: The authors well addressed the reviewers' comments and added enough data to complete the story with a focused angle of view. I suggest to publish the manuscript in Nature Communications.
1. Response by the authors: Reviewer #2 is gratefully acknowledged for her/his appreciation of the authors' efforts to revise the paper according to the Reviewers' suggestions.

Reviewer #3

1. Reviewer's comment: In the first round of reviews there were sever concerns about the novelty of the manuscript and several technical questions. I do not think that either of the aspects have been well answered. Especially as the author claim that the aim has changed. However, the claim that a preformed corona on nanoparticles change the biological fate is not novel.

1. Response by the authors: We thank the reviewer for his time and consideration; however, to the best of our knowledge, no previous study has mechanistically investigated the interactions of circulating leukocytes with corona-coated cationic, zwitterionic and anionic liposomes. We have thoroughly considered the reviewers' former comments, and exposed both bare liposomes and corona-coated liposomes to whole blood and probed their interactions with leukocytes.

2. Reviewer's comment: One of the technical questions was that it is not clear that all liposomes have been pelleted after incubation with blood plasma. The authors try to measure the the pelleting by determining the DLS signal after centrifugation. This is not a good method to use for several reasons. I give some examples. In most DLS instrument he auto attenuation function regulates the strength of the laser and the sensitivity of the detectors. This means that the setting is optimized for each reading and the concentration of the measured particles cannot be compared from sample to sample. If the laser and detector is locked it might be possible but the authors, as far as I have seen has not reported the settings. And also then for a specific setting larger particles would scatter more light and at that setting small particles may not be detected. As the authors themselves point out larger aggregates of liposomes and proteins pellets easily. Small aggregates, maybe with a single liposome would be much more difficult to detect with DLS especially in a high protein background.

2. Response by the authors: To address this point, which is similar to the point raised by Reviewer # 1, we used ultra-high-performance liquid chromatography electrospray ionization tandem mass spectrometry (UHPLC/ESI-MS/MS). UHPLC/ESI-MS/MS is a very powerful method for estimating very low concentration of lipids in biological samples (see references 65-67 in the revised version of the paper). After centrifugation, we quantified lipid concentration in the supernatant for each of the six experimental conditions (i.e. lipid species and HP percentage). Lipid concentration in the supernatant ranged between 26.5 ng/ μ l (DOTAP liposomes incubated in 50% HP) and 42.8 ng/ μ l (DOPC liposomes incubated in 5% HP). Considering the initial lipid concentration (1 mg/ml), this means that the percentage of lipid in the supernatant spanned between 2.6% and 4.3%. Then, the pellets were washed three times with PBS to remove unbound and loosely bound proteins (the 'soft corona') obtaining liposomes coated by the so-called 'hard corona'. To check whether washing steps reduced lipid amount in lipid-protein complexes, we quantified liposome concentration in the washing buffer after each of the three washing steps. This procedure allowed us to accurately estimate the percentage of lipid in lipid-protein complexes (i.e. in the pellet) with respect to bare liposomes (i.e. before exposure to HP). This percentage ranged between 95.6% (DOPC exposed to 5% HP) and 97.3% (DOTAP exposed to 50% HP).

Figure S1. Multiple reaction monitoring (MRM) transitions (A) and chromatograms (B) of DOTAP, DOPC and DOPG. (C) Calibration curves for DOTAP (red points), DOPC (green points) and DOPG (blue points) in the concentration range between 0.01 ng/μL and 3.5 ng/μL. Solid lines represent the best linear fits to the data. R² values were: 0.9985 (DOTAP), 0.9995 (DOPC) and 0.9991 (DOPG). (D) Liposome-protein complexes were isolated by centrifugation for 15 min at 14,000 rpm. Then, pellets were washed three times with PBS to remove unbound and loosely bound proteins (the ‘soft corona’) obtaining the so-called ‘hard corona’. Lipid concentration was calculated in the supernatant (i.e. after centrifugation) as well as after each of the three washing steps. When lipid concentration was out of the linear range (i.e. higher than 3.5 ng/μL), samples were pre-diluted. Dilution factors were considered to estimate lipid concentration. (E) Lipid percentage in lipid-protein complexes with respect to bare liposomes (i.e. before exposure to HP). Results are provided as average ± standard deviation of three replicates.

REVIEWERS' COMMENTS:

Reviewer #1 (Remarks to the Author):

We would like to thank the authors for taking the time and efforts to include our recommendations. We are happy with the revised version of the paper.

Reviewer #3 (Remarks to the Author):

I believe the measurements of the remaining lipids after centrifugation are now well enough described. I still don't think the novelty of the manuscript is high. There is always new particles and new conditions that can be tested but that doesn't automatically mean the data should be published in journals as nature communication. I have no further comments.